# Isolated Systolic Blood Pressure and Red-Complex Bacteria—A Risk for Generalized Periodontitis and Chronic Kidney Disease

**DOI:** 10.3390/microorganisms10010050

**Published:** 2021-12-27

**Authors:** Jaideep Mahendra, Plato Palathingal, Little Mahendra, Janani Muralidharan, Khalid J. Alzahrani, Mohammed Sayed, Maryam H. Mugri, Mohammad Almagbol, Saranya Varadarajan, Thodur Madapusi Balaji, Shilpa Bhandi, Sruthi Srinivasan, A. Thirumal Raj, Shankargouda Patil

**Affiliations:** 1Department of Periodontics, Meenakshi Ammal Dental College and Hospital, Chennai 600 095, Tamil Nadu, India; janani.harini718@gmail.com (J.M.); drsruthisvasan@gmail.com (S.S.); 2Department of Periodontics, Annoor Dental College, Ernakulam 686673, Kerala, India; Platoos@gmail.com; 3Research Department of Periodontics, Maktoum bin Hamdan Dental University, Dubai 213620, United Arab Emirates; littlemahendra24@gmail.com; 4Department of Clinical Laboratories Sciences, College of Applied Medical Sciences, Taif University, P.O. Box 11099, Taif 21944, Saudi Arabia; ak.jamaan@tu.edu.sa; 5Department of Prosthetic Dental Sciences, College of Dentistry, Jazan University, Jazan 45412, Saudi Arabia; drsayed203@gmail.com; 6Department of Maxillofacial Surgery and Diagnostic Sciences, College of Dentistry, Jazan University, Jazan 45412, Saudi Arabia; dr.mugri@gmail.com; 7Department of Community and Periodontics, Faculty of Dentistry, King Khalid University, Abha 62529, Saudi Arabia; malmagbol@kku.edu.sa; 8Department of Oral Pathology and Microbiology, Sri Venkateswara Dental College and Hospital, Chennai 600130, Tamil Nandu, India; vsaranya87@gmail.com (S.V.); thirumalraj666@gmail.com (A.T.R.); 9Tagore Dental College and Hospital, Chennai 600127, Tamil Nandu, India; tmbala81@gmail.com; 10Department of Restorative Dental Science, College of Dentistry, Jazan University, Jazan 45142, Saudi Arabia; shilpa.bhandi@gmail.com; 11Department of Maxillofacial Surgery and Diagnostic Sciences, Division of Oral Pathology, College of Dentistry, Jazan University, Jazan 45412, Saudi Arabia

**Keywords:** blood pressure, chronic, glomerular filtration rate, hypertension, isolated systolic blood pressure, kidney disease, males, periodontal disease, periodontitis, red-complex bacteria

## Abstract

Hypertension is a risk factor for generalized periodontitis (GP) and chronic kidney diseases (CKD). However, the role of isolated systolic blood pressure as one of the major risks for these inflammatory diseases has not been explored. Very limited studies exist identifying the red-complex bacteria in association with the isolated systolic blood pressure. Hence, the main objective of this study was to assess the isolated systolic blood pressure and the red-complex bacteria along with the demographic variables, periodontal parameters, and renal parameters in patients with generalized periodontitis and chronic kidney disease. One hundred twenty participants (age 30–70 years) were divided into four groups—Group C: control (systemically and periodontally healthy subjects), Group GP: generalized periodontitis, Group CKD: subjects with CKD with good periodontal health, Group CKD + GP: subjects with both generalized periodontitis and CKD. Demographic variables and periodontal parameters were measured and recorded. Blood pressure measurements and a detailed history and renal parameters such as serum creatinine, eGFR, and fasting blood sugar were recorded. The red-complex bacteria (RCB) were assessed in the subgingival plaque samples of all four groups using RT-PCR. Older participants (above 50 years) showed worse periodontal scores in the CKD + GP group along with elevated isolated systolic blood pressure, higher serum creatinine, and fasting blood sugar. eGFR was significantly decreased compared to the other groups. Bacterial counts were higher in the GP + CKD group, suggesting that they may be at a higher risk for generalized periodontitis and chronic kidney disease. Isolated systolic blood pressure (ISBP) and RCB were significantly correlated with the renal and periodontal parameters. A log-linear relationship exists between periodontal disease, CKD, RCB, and isolated systolic hypertension levels.

## 1. Introduction

Clinical hypertension and impaired regulation of arterial blood pressure and are two major systemic conditions associated with inflammation and immune dysfunction [1,2]. It is a prevalent risk factor for cardiovascular disease and the progression of chronic kidney disease. Hypertensive cardiovascular disease has risen to become the ninth leading cause of death worldwide. Systolic blood pressure ranging from 115 mm/Hg to 180 mmHg was found to be associated with an approximately two-fold increase in the risk of kidney failure and mortality [1,3] Previous epidemiological studies have indicated that 45% of the adult population in the United States [4] and 30–45% in European countries are hypertensive [5]. Hypertension-related mortality in the United States was estimated to be half a million in 2018 [6].

Chronic kidney disease is a complex multifactorial pathogenic disorder. The main risk factors are reduced physical activity, increased body mass index (BMI), elevated blood pressure and increased blood cholesterol levels. Periodontal disease has recently been identified as an important potential risk factor for CKD [7]. Studies revealed that CKD affects teeth, oral mucosa, periodontium, salivary glands, and tongue, resulting in a negative effect on the oral health status of the patient. Increased levels of plaque have been reported for hemodialysis (HD) patients. There is a higher prevalence of periodontal disease in CKD patients [8]. Hypertension plays a key role in the development and progression of chronic kidney disease (CKD). It is a major complication for uncontrolled hypertension can act as both the cause and the effect [9]. This interaction between hypertension and CKD poses an increased risk for adverse cardiovascular and cerebrovascular events [9]. Resistant hypertension is commonly seen in patients with CKD. The factors involved in the pathogenic mechanism of CKD and hypertension include sodium dysregulation, alterations in the renin-angiotensin-aldosterone axis, and increased sympathetic nervous system activity [8,9].

Periodontitis is a chronic inflammatory disease associated with gram-negative bacteria leading to the destruction of the tooth-supporting structures and ultimately tooth loss. It affects nearly one in two adult individuals worldwide [10]. The red-complex bacteria (RCB), namely *Tanerella forsythia* (*Tf*), *Treponema denticola* (*Td*), and *Porphyromonas gingivalis* (*Pg*), are most commonly seen in the diseased periodontitis sites of the patients [11]. The association of these bacteria with CKD and GP is not well explored. Poor oral hygiene and oral dysbiosis with higher periodontal bacterial count have shown to be significantly correlated with the severity of chronic kidney disease. Periodontal pathogens may be linked to distant organ infection caused due to the spread of bacteria, antigens, endotoxins, and inflammatory cytokines through the circulatory system. Diseased periodontal tissue being a source of chronic systemic inflammation, generalized periodontitis may be a possible risk factor for CKD [11].

Periodontitis and hypertension share common risk factors of older age, male gender, smoking, overweight/obesity, diabetes, and low socioeconomic status. Recent studies have revealed that the association between periodontitis and hypertension extends beyond the sharing of common risk factors. Epidemiological studies reveal periodontitis to be an independent risk factor for hypertension [11,12,13].

When considering the association of complex multifactorial diseases, two distinct scenarios are possible: a particular genetic background predisposes to the two diseases, or one complex disease predisposes to the other, or both predispose to each other. CKD and periodontitis are two diseases, with hypertension being a common risk factor. Few studies have explored the association between isolated systolic blood pressure and the RCB in generalized periodontitis and systemic disorders such as chronic kidney disease [12,13]. Keeping the aforementioned evidence in mind, we hypothesized that isolated systolic blood pressure might act as a risk factor for chronic periodontitis with CKD in association with the presence of red-complex bacteria. Hence, this study aimed to evaluate the demographic variables, periodontal parameters, presence of red-complex bacteria, renal parameters, and isolated systolic blood pressure in patients with generalized periodontitis and chronic kidney disease to assess their interrelationship. Our primary objective was to assess the isolated systolic blood pressure along with the assessment of the demographic variables, periodontal parameters, and renal parameters in patients with generalized periodontitis and chronic kidney disease and compare them with the other study groups. The secondary objective was to assess the presence of red-complex bacteria and explore the association with the other variables.

## 2. Materials and Methods

The study was conducted between January 2018 to February 2020 in Thrissur, Kerala, India. Two hundred patients were recruited from PSM College of Dental Science and Research, Thrissur, India. Both male and female participants aged 30 to 70 years were included in the study. Study participants who were unwilling or did not meet the inclusion criteria were excluded. Subjects who had periodontal treatment in the past 6 months, with smoking/tobacco/alcohol habits, patients with carcinoma, sarcoidosis, immunosuppressive conditions, rheumatoid arthritis, and tuberculosis were excluded from the study.

One hundred twenty subjects were selected as the study population. This study was approved by Institutional Ethical Review Board, PSM College of Dental Science and Research, Thrissur, India (Institutional ethical committee Protocol no: PSMDC/IEC/01/2015). The study was conducted according to the guidelines of the Helsinki Declaration of 1975, revised in 2013. The study protocol was explained in detail to the patients, and written informed consent was obtained from all participants.

The study subjects were divided into four equal groups:

Group C: Control group of 30 subjects who were systemically and periodontally healthy;

Group GP: 30 subjects with Generalized Periodontitis (group GP) with at least 10 remaining natural teeth and generalized periodontitis;

Group CKD: 30 patients with chronic kidney disease alone. Periodontally healthy CKD patients who were not on CKD stage 5 and dialysis were selected. Patients without CKD presented an estimated glomerular filtration rate >90 mL/min, estimated according to CKD-EPI equation 2009 [14];

Group CKD + GP: 30 subjects with both and periodontitis (CKD + GP) (Figure 1). Patients with periodontal disease fulfilling the definition given by AAP in 2017 with stage 2 grade B periodontitis, also suffering from CKD except stage 5 and on dialysis, were selected.

The parameters assessed in the study included the demographic, periodontal, and renal parameters.

### 2.1. Demographic Variables

Demographic variables included age (30 to 70 years), body mass index (BMI, kg/m^2^), and sex.

### 2.2. Periodontal Parameters

The periodontal assessment was performed by a skilled and experienced investigator. The periodontal examination was done using a William periodontal probe. Periodontal parameters included: (1) Plaque Index (PI) [15], (2) Gingival Index (GI) [16], (3) Probing Pocket Depth (PPD), (4) Clinical Attachment Level (CAL) [17], (5) Percentage of sites with pockets. The measurements were documented to the closest millimeter. PPD and clinical attachment loss (mm) were calculated as a mean of all sites of the entire dentition. PI was recorded using the index proposed by Loe and Silness (1963), and GI was recorded based on the index described by Sillness and Loe (1964). These indices were calculated as a mean of four sites (mesiobuccal, midbuccal, distobuccal, and lingual/palatal) for the entire dentition. The percentage of sites with pockets was calculated based on the sites with a probing pocket depth ≥ 5 mm.

Criteria for diagnosis of the generalized periodontitis: Participants were diagnosed with generalized periodontitis if they had at least 30% of tooth sites: (1) with probing depth (PD) ≥ 5 mm at one or more sites; (2) CAL ≥ 3 mm at the same site with positive bleeding on probing, with stage II grade B periodontitis patients, according to the recent 2017 World Workshop Classification of Periodontal and Peri-implant diseases were selected. For periodontally healthy subjects, PPD ≤ 3 mm in every site with no bleeding on probing was considered.

### 2.3. Renal Parameters

Venous blood samples (5 mL) were collected in the morning after a 12 h fasting period. The samples were transported to the laboratory for biochemical analysis. The serum creatinine and fasting blood sugar were determined by an automated method, using a semi-automated biochemical analyzer (Hospitex, Master T, Via Massimo D’Antona 17 50019–Sesto Fiorentino (Florence)–Italy). eGFR was calculated based on serum creatinine values using CKD—EPI equation 2009 [14].

### 2.4. Assessment of Blood Pressure

At the hospital visit, the participants were asked to complete questionnaires. A detailed case history, including a history of past medications and dental history, was taken, followed by a physical and oral examination. All relevant parameters were recorded. Before blood pressure examination, the patients were advised to take a rest for five minutes. Blood pressure (SBP and DBP) measurements were documented twice on the right arm using a stethoscope and a sphygmomanometer in a hospital room away from excessive workplace noise. Systolic and diastolic blood pressures were defined using the first and fifth Korotkoff sounds, respectively. Measurements were rounded upward to the nearest 2 mm Hg and recorded. Previous diagnosis and/or treatment of hypertension or high blood pressure by a physician was assessed via a questionnaire.

Hypertension was defined as taking self-reported physician diagnosis or treatment for hypertension, blood pressure-lowering medication on medication inventory, having systolic blood pressure ≥140, or diastolic blood pressure ≥90 [18]. Secondary outcomes of three-category severity of hypertension, was taken into consideration which was based on SBP and DBP cutoff points suggested by the JNC7 report [19] (Stage 0 if SBP < 140 mmHg and DBP < 90 mmHg; Stage 1 if 140 mmHg ≤ SBP < 160 mmHg or 90 mmHg ≤ DBP < 100 mmHg; Stage 2 if SBP ≥ 160 mmHg or DBP ≥ 100 mmHg). The hypertensive SBP and DBP categories included those individuals who were under medication for hypertension.

### 2.5. Assessment of Potential Confounders

The study analyzed the potential confounders such as age, sex, BMI, and obesity as a part of demographic variables. Diabetes mellitus associated with CKD, generalized periodontitis and hypertension were also taken into account [19].

### 2.6. Red-Complex Bacteria (RCB)

Molecular Analysis: Red-complex bacteria: *Tanerella forsythia* (*Tf*), *Treponema denticola* (*Td*), and *Porphyromonas gingivalis* (*Pg*) were quantified in subgingival plaque samples using real-time polymerase chain reaction. The diseased area was first isolated with the sterile cotton roles, and the supragingival plaque and calculus were removed. Subgingival plaque was then collected using periodontal curettes from the sites with the deepest periodontal pockets present in the entire dentition and transferred to the sterile vial for further molecular analysis.

DNA isolation from the subgingival plaque: A laboratory phenol extraction procedure was used to obtain plaque DNA (1987, Chomczynski and Sacchi). One hundred milliliters of buffer containing 10% sodium dodecyl sulphate solution (SDS) and 5 mL of Proteinase-K were used to incubate the samples for an hour at 37 °C. After incubation, 1 mL mixture of phenol and chloroform were added to the samples and centrifuged for 10 min at 10,000 rpm. The supernatant was collected, and the mixture was extracted. The supernatant was stored for 2 h at −200 °C before being treated with 0.5 mL isopropanol solution. The final product was centrifuged for 15 min at 10,000 rpm. The final fractions were rinsed in 0.5 mL of 70% alcohol and were liquified in 0.2 mL sterile milli-Q water. The quantity of DNA was measured with a spectrophotometer at a wavelength of 260–280 nm. The DNA was stored at −200 °C for subsequent examinations.

#### PCR Procedure for the Identification of Red-Complex Bacteria

Red-complex bacteria: The primers for red-complex bacteria were developed using 16S rRNA as a reference and were generated using NCBI-BLAST (Table 1). In RT-PCR, SYBR Green^‖^ dye was utilized to identify microbial DNA. Species-specific primers were utilized for the quantification of Pg, Tf, and Td. ^§^ Melt curve analysis was used to assess the presence of nonspecific products, numerous amplicons, and impurities in each sample. CT technique (comparative cycle threshold units) was used to determine the number of a gene. The CT value represented the number of bacteria in the red complex. The comparative estimation of bacteria was obtained using the normal amplification curve produced from the typical genomic DNA. Lower CT values represented higher levels of red-complex bacteria in the subgingival plaque samples.

The PCR results were seen using a 2% agarose gel electrophoresis with 100 base pair DNA. It was observed using a UV transilluminator in the gel documentation system at a wavelength of 260 nm. A solid PCR plate with 96 wells was used to prepare the mix, along with translucent seal plates. The plate edges and corners were sealed to prevent evaporation-related artifacts. The products were amplified using a real-time thermocycler. The real-time instrument’s SYBR Green channel was employed for quantification utilizing Luna Master Mix (BIORAD-CFX100).

### 2.7. Statistical Analysis

The statistical analysis was performed using SPSS, v.17 in Microsoft Windows. The data were normally distributed. In each of the four groups, the mean ± SD was calculated for all the recorded parameters, and the level of significance was estimated. To measure metric data under normal distribution, Kolmogrov–Smirnov and Shapiro tests were used. For intergroup comparisons, a chi-square test and ANOVA were used for all parameters. Pearson correlation analysis was done to correlate all the variables in all four study groups and the red-complex bacteria with all periodontal and renal parameters. A *p*-value ˂ 0.05 was considered statistically significant. The ternary SBP and DBP variables adopted the diagnostic criteria as suggested by the JNC7 report [19].

## 3. Results

Intergroup comparisons were carried out between all the variables and the control group. On comparing demographic variables between the various groups, mean age was highest in the CKD + GP groups, and the difference was statistically significant among the groups with *p* = 0.0001 (Table 2). Several subjects with ages less than 50 and more than 50 reached a level of statistical significance in all four groups. There were no statistically significant differences observed in gender and BMI variables in all the four groups (Table 2).

A fewer number of teeth were present in the GP, CKD, and CKD + GP groups compared to the C group; however, the result was found to be statistically insignificant. The plaque index and gingival index were found to be significantly higher in the CKD + GP group and GP group than compared to the other groups (*p* = 0.001). Mean probing pocket depth was highest in the CKD + GP group, followed by the GP group. The percentage of sites with periodontal pockets was higher in the CKD + GP group, with a significant difference when compared to the GP group (*p*-value = 0.001). The mean CAL was highest in the CKD + GP group compared to the GP group showing a statistically significant difference (Table 3).

Creatinine values were significantly elevated, and eGFR values were lowered in CKD and CKD + GP groups compared to the other groups (*p* = 0.001). The frequency of diabetic individuals was higher in the CKD + GP group, CKD, and GP group than in the C group (Table 4).

Mean systolic blood pressure values were significantly higher in the CKD + GP, CKD, GP groups compared to the C group. The frequency of hypertensive subjects was higher in CKD and CKD + GP than in the GP group. Stage 2 hypertension was higher in CKD + GP, GP, and CKD group than in the C group. Stage 1 hypertension was significantly higher in the CKD group than GP and CKD + GP group. The frequency of subjects with SBP > 140 mmHg was statistically significantly higher in the CKD + GP group as compared to the CKD and GP groups. SBP >120–139 mmHg was higher in the CKD + GP group when compared to other groups. Isolated SBP values were significantly increased in the CKD + GP group compared to the CKD and GP (*p* = 0.0001) (Table 5).

On comparison of red-complex bacteria, CT values in C, GP, CKD, and GP + CKD group showed lower CT values in the GP + CKD group, indicating increased levels of bacteria. The results were statistically significant (Table 6).

Overall Systolic Blood Pressure showed a significant correlation with all the variables of age, missing teeth, creatinine values, eGFR, PI score, and mean CAL (Table 7).

Pearson’s correlation of the red-complex bacteria with the periodontal parameters and renal parameters showed a significant correlation of *Pg*, *Td*, and *Tf* with PI, GI, PPD, CAL, number of teeth present, and eGFR (Table 8).

## 4. Discussion

Hypertension is a common risk factor for both chronic kidney disease and generalized periodontitis [19]. Many etiological factors link periodontal disease, CKD, and hypertension. Periodontal disease can cause a rise in blood pressure leading to the activation of the immune system. This can lead to increased oxidative stress, activation of the sympathetic nervous system, autoimmune reactivity, and increased inflammatory cytokine levels [19]. Hypertension is also a risk factor for systemic diseases such as diabetes, CKD, and cardiovascular disease and is associated with higher morbidity and mortality [20].

This study set out to examine the relationship between isolated systolic blood pressure, generalized periodontitis, and chronic kidney disease in correlation with demographic, periodontal variables, and renal parameters taking the confounding factors into account. Our findings showed that the CKD + GP group had significantly older participants (above 50 yrs) as compared to the remaining groups (Table 2). This result is similar to those observed by Romandini et al. [21], Ahn et al. [18], and Sharma et al. [22]. Age can be a predictive risk factor for the development of chronic inflammatory diseases due to its influence on the vasculature, blood supply, oxidative stress, and normal tissue homeostasis. Advanced age is associated with the marked release of proinflammatory cytokines, which may impair normal renal function [21,23]. There is an increased incidence of chronic renal diseases in older individuals. This can raise the risk of their developing severe periodontal diseases [21]. There were no statistically significant differences observed in gender in all four groups. This confirms the findings of Sharma P et al. [22], who observed that males and females are equally affected with CKD, hypertension, and periodontal disease.

The Body Mass Index (BMI) was similar in all four study groups in our study. This result reflects those of Caula et al. [24] and Messier et al. [25] (Table 2). BMI is an important predictor of cholesterol levels in obese individuals. A high BMI is related to increased blood pressure and triglyceride levels. This can influence the inflammatory pathways of the body. These conditions predispose individuals to systemic inflammatory diseases such as CKD and periodontal disease [26].

Consistent with earlier literature [22], we found that the CKD + GP, CKD, and GP groups had the lowest number of teeth present compared to the C group (Table 2). The number of teeth present is a predictor of past periodontal destruction or other dental problems. The incidence of tooth loss is higher in individuals with chronic inflammatory diseases such as generalized periodontitis and chronic kidney disease [27]. Periopathogenic bacteria and inflammatory mediators cause tissue destruction, which can compromise the integrity of the supporting periodontal apparatus and eventually lead to tooth loss. These noxious agents also may traverse the systemic circulation and colonize the renal tissues resulting in tissue damage. This may explain the higher incidence of tooth loss in both inflammatory diseases [27,28]

CKD + GP group showed higher plaque index (PI) and gingival index (MGI) scores compared to the other groups (Table 3). This corroborates earlier research by Baioni et al. [8], who demonstrated that persons who suffered from renal disease and periodontitis had higher index scores and comparatively worse oral hygiene. The plaque index and gingival index scores aid in the assessment of the oral hygiene status, plaque burden, and inflammatory status of the periodontium. Poor oral hygiene is a risk factor that contributes to gingival inflammation, which may lead to the systemic spread of inflammatory mediators, causing organ damage. Similar to the results obtained by Davidovich et al. we observed a lower mean PI and GI score in the CKD group compared to the CKD + GP group [29].

The percentage of periodontal pockets, mean PPD, and CAL values were statistically significantly higher in the CKD + GP group compared to the other groups (Table 3). These results are in accord with previous research by Kshirsagar et al. [30] and Grubbs et al. [23]. PPD and CAL express the severity of periodontal disease. Worsening periodontal status would increase the likelihood of tooth loss and be a risk for blood pressure dysregulation [31,32]. Vidal et al. reported that hypertensive patients had a higher proportion of sites with dental plaque, gingival bleeding, and an increased number and proportion of sites with clinical attachment loss (6 mm or more) than non-hypertensive patients [33]. An earlier study examined the relationship between the levels of periodontal bacteria and hypertension and established a direct association between the two diseases [34]. Periodontitis and hypertension are chronic– their longer duration and bidirectional nature contribute to worse disease outcomes [18].

Chronic kidney disease is associated with an increased prevalence of cardiovascular diseases in older, hypertensive and diabetic patients [35]. Inflammation and vascular damage may influence the progression of renal disease [35]. Creatinine is an important marker to measure renal function. Most serum creatinine is derived from skeletal muscle as a metabolite of creatine [27]. The mean creatinine values were higher in CKD groups as compared to the non-CKD (Table 4). The mean creatinine value in the present study in the CKD and GP groups was consistent with the results obtained by Caula et al. and Shimazaki et al. [24,27].

We observed a lower range and mean of eGFR value in CKD groups compared to the other groups. This matches the results of Chen et al. 2011 [36] (Table 4). GFR decline demonstrates the effects of periodontal disease on CKD progression. CKD is diagnosed when eGFR is less than 90 mL/min. This means the individual has mild to severe kidney function decline. Chronic renal failure is associated with marked disturbances of bone structure and metabolism, which may have an impact on periodontal disease [29]. Severe periodontal disease is associated with an increased glomerular filtration rate [37].

The number of diabetic individuals was higher in the CKD + GP group compared to the other three groups. These results mirror the findings of Sharma P et al. [22] and Almeida S et al. [38] (Table 4). A chronic inflammatory response due to periodontal inflammation may increase peripheral insulin resistance and predispose patients to renal disease [22]. Periodontal disease may influence the development of renal insufficiency in diabetic individuals.

CKD + GP group showed higher systolic blood pressure (SBP) compared to the other groups (Table 4). Periodontal pathogens may affect SBP indirectly through the increase in inflammatory substances in the circulation and directly at the endothelial wall level. This is the first study to use objective measures to examine the importance of isolated systolic blood pressure in periodontal disease and its association with chronic kidney disease.

The percentage of individuals in normal, prehypertension, and hypertension stages were lower in C and GP groups as compared to the other groups (Table 5). This was in keeping with previous studies by Romandini et al. [21] and Ahn et al. [18]. CKD + GP group showed significantly high values of isolated systolic blood pressure. The number of hypertensive patients was highest in the CKD + GP, which supports earlier clinical observations by Lertpimonachai et al. [20] and Tumanyan et al. [1]. The association between periodontitis and hypertension is pivotal because the development and progression of cardiovascular disease are influenced by blood pressure. The identification of adjustable risk factors for the initiation and progression of hypertension has critical significance at a global level. Hypertension continues to be a major cause of morbidity and mortality and exacerbates other medical conditions [19]. Inflammation that is present systemically can also affect blood pressure control in elderly people [39,40]. Morita et al. demonstrated that the presence of periodontal pockets increased the risk for hypertension [41]. Holmlund et al. observed that there is a progressive linear trend between the severity of periodontal disease and self-reported treatment for hypertension [42].

CKD + GP group showed higher counts of red-complex bacteria, with lower CT values correlating to higher bacterial counts (Table 6). Kshirsagar et al. [30] also reported an increase in the bacterial count in CKD patients. Fischer et al. [39] compared the periodontopathogenic bacteria between CKD and non-CKD groups and found that CKD patients with generalized periodontitis having higher bacterial counts. Periodontal bacteria and their inflammatory mediators can disseminate to the blood vessels and can reach distant sites causing the inflammatory reaction, thereby increasing the oxidative stress, which may exert a profound impact on the blood vessels, leading to high blood pressure, specifically isolated systolic blood pressure, thereby posing to be risk factors for both GP and CKD. Moreover, the antibody response to periodontal bacteria creates an inflammatory status, which predisposes to CKD. Additionally, in CKD patients, the uremic environment creates an alkaline pH which also favors the multiplication of proteolytic periodontal pathogens. Hypertensive patients with CKD may be at an increased risk for infections because of their immunocompromised state [39,42].

Pearson’s correlation coefficient analysis was used to correlate SBP with the demographic, periodontal, and renal parameters (Table 7). Systolic blood pressure correlated significantly with variables such as age, Mean Clinical Attachment loss, creatinine, and estimated glomerular filtration rate (*p* ≤ 0.05). Periodontitis-induced systemic inflammation may contribute to the severity of CKD [34]. Periodontal disease and systemic inflammation are associated with an increased risk of hypertension, thus establishing a bidirectional relationship [43]. Diseases with low-grade inflammation, such as diabetes and hypertension, are commonly associated with CKD [44]. Metabolic syndrome, which includes a combination of hypertension, hypercholesterolemia, hyperlipidemia, is associated with periodontitis [45,46]. Inflammation could be a risk for hypertension and periodontitis [47]. Both diseases are chronic inflammatory diseases that are linked to oxidative damage and proinflammatory cytokines [48]. These inflammatory molecules in systemic circulation could be a common risk for hypertension and periodontitis [18]. Endothelial dysfunction induced by high blood pressure and the resulting reduction in blood flow to periodontal tissues may heighten the susceptibility of periodontal tissues to inflammation [49].

We correlated the periodontal bacteria present with other variables and found them to be statistically significant. An increased count of red-complex bacteria was correlated with the periodontal parameters, raised serum creatinine levels, and lower eGFR values (Table 8). Persistent low-level bacteremia in patients with periodontitis leads to bacterial infiltration of the renal glomerulus. Here they are filtered out and invade the endothelium, mesangial cells, and matrix. This creates fluctuations in the serum creatinine and eGFR values, eventually resulting in decreased renal function [30].

The sampling strategy and the data collection methods used in this study allowed minimization of selection or information bias. Confounding variables were analyzed to make the results credible. As the study participants were not asked to stop taking their prescribed medications before the assessment, the blood pressure measurements obtained from those on antihypertensive medication reflect the genuine degree of blood pressure control. Additionally, the present study had data on multiple covariates that allowed the control of potentially confounding factors. Hence, the results of the present study are reliable to test the hypothesis that increased isolated systolic blood pressure is associated with periodontitis and chronic kidney disease.

A limitation of our study is its cross-sectional design. This temporally limits the assessment of periodontal disease and hypertension occurrence. Our sample size was limited, which could hamper the precise estimation of the true direction and degree of association of the diseases. This is an important issue for future research. Long-term longitudinal studies should be done to corroborate these findings and to develop a fuller picture of the association between red-complex bacteria, isolated systolic hypertension, and chronic kidney disease.

## 5. Conclusions

Based on our findings, we concluded that isolated systolic blood pressure and RCB counts were significantly high in chronic kidney disease patients with generalized stage II grade B periodontitis. Hence isolated systolic blood pressure and the presence of red-complex bacteria may pose the potential risk factors for generalized periodontitis and chronic kidney disease, which may be associated with adverse outcomes in older adults. Thus the control of isolated systolic hypertension and reduction in the red complex bacterial counts becomes essential to limit the risk for generalized periodontitis and chronic kidney disease.

## Figures and Tables

**Figure 1 microorganisms-10-00050-f001:**
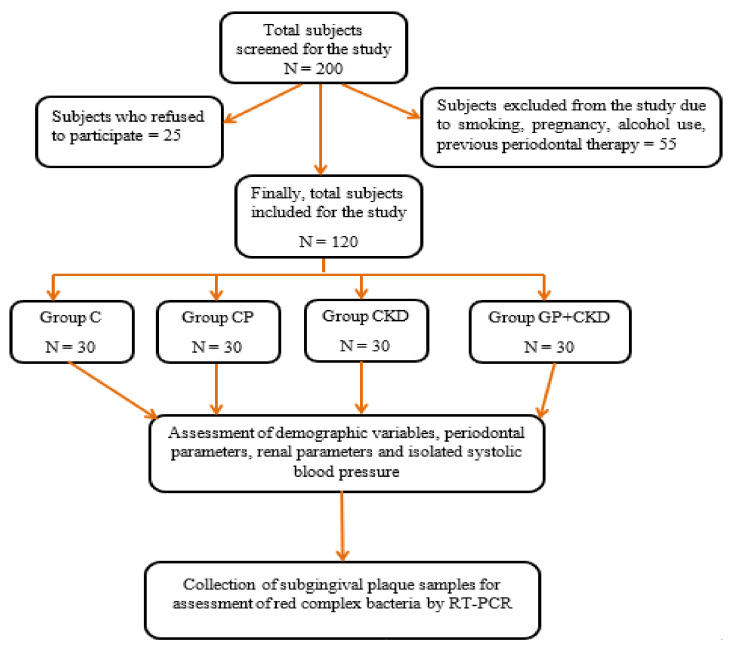
Flow chart of the study design.

**Table 1 microorganisms-10-00050-t001:** Primer sequences for assessment of red-complex bacteria using RT-PCR.

Bacteria	Primer Sequences
Tf (Forward primer)	GCGTATGTAACCTGCCCGCA
Tf (Reverse primer)	TGCTTCAGTGTCAGTTATACC
Pg (Forward primer)	AGGCAGCTTGCCATACTGC
Pg (Reverse primer)	ACTGTTAGCAACTACCGATGT
Td (Forward primer)	TAATACCGAATGTGCTCATTTACAT
Td (Reverse primer)	TCAAAGAAGCATTCCCTCTTCTTCTTA

A: Adenine; G: Guanine; C: cytosine; T: thymine; Tf: *Tannerella forsythia*; Pg: *Porphyromonas gingivalis*; Td: *Treponema denticola*.

**Table 2 microorganisms-10-00050-t002:** Intergroup comparison of the demographic parameters assessed among C, GP, CKD, and CKD + GP groups.

Parameters	C	GP	CKD	CKD + GP	*p*-Value
N	30	30	30	30	
Mean Age (years)	37.63 ± 10.26	54.03 ± 9.10	59.27 ± 10.90	61.47 ± 10.99	0.0001 *
Age < 50 years	22	9	5	4	0.001 *
Age > 50 years	8	21	25	26	0.001 *
Male	14	13	17	19	0.39
Female	16	17	13	11	0.39
BMI	25.70 ± 3.48	26.39 ± 4.39	25.81 ± 3.79	26.48 ± 4.81	0.846
Obesity (kg)	2	6	2	5	0.274

C: Controls; GP: Generalized Periodontitis; CKD: chronic kidney disease; * significant (ANOVA), *p* < 0.05.

**Table 3 microorganisms-10-00050-t003:** Intergroup comparison of periodontal parameters assessed among C, GP, CKD, and CKD + GP Groups.

Parameters	C	GP	CKD	CKD + GP	*p*-Value
N	30	30	30	30	
Teeth Present	27 ± 1.48	25.53 ± 3.36	25.73 ± 3.03	25.37 ± 4.06	0.173
Plaque Index	0.80 ± 0.29	1.79 ± 0.22	0.88 ± 0.29	2.27 ± 0.23	0.001 *
Gingival Index	0.91 ± 0.34	1.93 ± 0.35	1.05 ± 0.30	2.42 ± 0.18	0.001 *
Mean PPD (mm)	1.33 ± 0.17	2.77 ± 0.27	1.29 ± 0.09	3.18 ± 0.24	0.001 *
% of sites with Pockets	0	12.04 ± 9.29	0	21.87 ± 7.49	0.001 *
Mean CAL (mm)	0	0.72 ± 0.26	0	1.14 ± 0.48	0.002 *

C: Controls; GP: Generalized Periodontitis; CKD: chronic kidney disease; * significant (Chi-square test, ANOVA), *p* < 0.05.

**Table 4 microorganisms-10-00050-t004:** Intergroup comparison of biochemical parameters assessed among C, GP, CKD, and CKD + GP Groups.

Parameters	C	GP	CKD	CKD + GP	*p*-Value
N	30	30	30	30	
CREATININE (mg/dl)	0.80 ± 0.14	0.72 ± 0.11	1.21 ± 0.43	1.08 ± 0.19	0.001 *
eGFR (mL/min/m^2^)	107.67 ± 14.81	101 ± 7.10	64.77 ± 18.94	68.43 ± 12.45	0.001 *
DIABETES	0	9	7	20	0.001 *

C: Controls; GP: generalized periodontitis; CKD: chronic kidney disease; * significant (Chi-square test, ANOVA), *p* < 0.05.

**Table 5 microorganisms-10-00050-t005:** Intergroup comparison of hypertensive variables among C, GP, CKD, and CKD + GP Groups.

Parameters	C	GP	CKD	CKD + GP	*p*-Value
N	30	30	30	30	
SBP mmHg	113.93 ± 8.48	125.27 ± 19.80	130.73 ± 14.31	132.2 ± 21.80	0.0001 *
Normal Bood Pressure mmHg	19	9	5	4	0.0001 *
HTN (Prehypertension) 110–120 mmHg	11	10	10	16	0.001 *
HTN (STAGE 1) 120–140 mmHg	0	6	11	4	0.002 *
HTN (Stage 2) >140 mmHg	0	5	4	6	0.003 *

C: Controls; GP: generalized periodontitis; CKD: chronic kidney disease; HTN: hypertension; * significant (Chi-square test, ANOVA), *p* < 0.05.

**Table 6 microorganisms-10-00050-t006:** Intergroup comparison of CT values of red-complex bacteria among C, GP, CKD, and CKD + GP Groups.

Variable	C	GP	CKD	CKD + GP	*p*-Value
N	30	30	30	30	
Pg	29.77 + 1.20	25.99 + 2.09	27 + 1.54	24.33 + 2.39	0.001 *
Td	28.58 + 1.28	26.1 + 1.67	27.10 + 1.72	25.94 + 1.01	0.0001 *
Tf	31.33 + 1.27	29.52 + 2.46	29.30 + 1.60	27.90 + 2.02	0.002 *

C: Controls; GP: generalized periodontitis; CKD: chronic kidney disease; Tf: *Tannerella forsythia*; Pg: *Porphyromonas gingivalis*; Td: *Treponema denticola*; * significant (Pearson’s correlation), *p* < 0.05.

**Table 7 microorganisms-10-00050-t007:** Overall Pearson’s correlation of SBP with demographic, periodontal, and renal parameters.

Correlations	Systolic Blood Pressure (mm Hg)	*p*-Value
	Pearson Correlation	
Age In Completed Years	0.472	0.002 **
Body Mass Index (BMI)	0.14	0.13
Number Of Teeth Missing	0.481	0.00 **
Creatinine Value—mg/dl	0.256	0.01 **
Estimated Glomerular Filtration Rate—Value (CKD-Epi 2009 Equation)—(mL/min/m^2^)	−0.369	0.001 **
Plaque Index Score	0.201	0.03 *
Gingival Index Score	0.10	0.26
Mean Probing Pocket Depth—mm	0.17	0.07
Mean Clinical Attachment Loss—mm	0.269	0.001 **

** highly significant, * significant (Pearson’s correlation) *p* < 0.05.

**Table 8 microorganisms-10-00050-t008:** Overall Pearson’s correlation of red-complex bacteria (RCB) with periodontal and renal parameters.

	*Porphyromonas* *gingivalis* CT Value		*Treponema denticola* CT Value		*Tannerella forsythia* CT Value	
	Pearson Correlation	*p*-Value	Pearson Correlation	*p*-Value	Pearson Correlation	*p*-Value
Plaque index (PI)	−0.616	0.00 **	−0.455	0.00 **	−0.433	0.001 **
Gingival index (GI)	−0.571	0.00 **	−0.467	0.00 **	−0.432	0.002 **
Mean probing pocket depth (PPD) mm	−0.567	0.00 **	−0.473	0.00 **	−0.396	0.003 **
Mean clinical attachment loss (CAL)—mm	−0.504	0.00 **	−0.407	0.00 **	−0.361	0.001 **
Number of teeth present	0.226	0.013 *	0.049	0.594	0.11	0.23
eGFR (CKD-EPI 2009 equation)—(mL/min/m^2^)	0.479	0.00 **	0.271	0.003 **	0.473	0.001 **

** highly significant, * significant (Pearson’s correlation), *p* < 0.05.

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
