# Peer review of "Isolated Systolic Blood Pressure and Red-Complex Bacteria—A Risk for Generalized Periodontitis and Chronic Kidney Disease"

_microorganisms, 2021, doi:10.3390/microorganisms10010050_

Round 1

Reviewer 1 Report

The authors examine in their study whether islotad systolic hypertension and red complex bacteria are putative risk factors for both, periodontitis and CKD. The topic is not uninteresting, but I have some comments that should be considered during the revision of the manuscript.

Major concerns

Abstract

A clear objective for the study should be formulated.

Introduction

At the end of the introduction, a study hypothesis should be formulated as well as main and secondary objectives of the study.

Methods

The determination of the periodontal parameters such as PPD, CAL, plaque and bleeding index should be precisely determined, e.g. how many sites per tooth were measured or what the meaning of percentage sites with pockets etc..

When was a patient diagnosed as a parodontal healthy and when he had a generalized periodontitis? For these definitiones, the new classification of 2017 should be considered.

How and which teeth, the subgingival plaque samples were taken for the RT-PCR?

How were the CT values calculated?

Statistical analyses

I miss that metric variables were tested for normal distribution. If normal distribution missing, parameter-free tests would have to be used.

The influence of gender, age, BMI, high blood pressure on the target variables periodontitis and CKD must have been tested in a multivariate analysis (e.g., logistical regression).

Results

It is unclear whether all groups were compared with each other or each group individually with the control group.

Discussion and Conclusion

A statement should be issued whether Red Complex Bacteria and hypertension are a risk factor for generalized periodontitis and CKD.

Minor comments

A native speaker should edit the entire manuscript. Abbreviations should be explained in the text during their first use and then always applied.

The outline of the tables should be improved. For example, the unit of measurement for each variable should be specified. The spelling of the P values should be unified. It must emerge from the tables which groups were compared with each other and which statistical procedure was applied.

  1. forsythensis improve in T. forsythia.

mMhg improve in mmHg

Author Response

Reply to Reviewer’s Comments

Reviewer 1:

The authors examine in their study whether isolated systolic hypertension and red complex bacteria are putative risk factors for both, periodontitis and CKD. The topic is not uninteresting, but I have some comments that should be considered during the revision of the manuscript.

Major concerns

Abstract

Q1: A clear objective for the study should be formulated.

Answer: The objective of the study has been included in the abstract section of the manuscript and highlighted as per the reviewer’s suggestions. (Lines:37-40 )

Introduction

Q2: At the end of the introduction, a study hypothesis should be formulated as well as main and secondary objectives of the study.

Answer: Study hypothesis along with the main and secondary objectives of the study have been included in the introduction section of the manuscript as per the reviewer’s suggestions. (Lines:106-108 )

Methods

Q3: The determination of the periodontal parameters such as PPD, CAL, plaque and bleeding index should be precisely determined, e.g how many sites per tooth were measured or what the meaning of percentage sites with pockets etc. When was a patient diagnosed as a periodontally healthy and when he had a generalized periodontitis? For these definitions, the new classification of 2017 should be considered.

Answer: The number of sites examined for PPD, CAL, PI and GI have been included. The criteria for the diagnosis for periodontitis and periodontally healthy subjects according to the recent 2017 classification has also been included in the methods section of the manuscript as per the reviewer’s suggestions. (Lines: 156-165)

Q4: How and which teeth, the subgingival plaque samples were taken for the RT-PCR?

Answer: The method for the subgingival plaque collection has been incorporated into the material and method section of the manuscript as per the reviewer’s valuable suggestions. (Lines:200-204)

Q5: How were the CT values calculated?

Answer: The method used to calculate the CT values has been explained in the manuscript as per the valuable suggestions. (Lines:225-229)

Statistical analyses

Q6: I miss that metric variable were tested for normal distribution. If normal distribution missing, parameter-free tests would have to be used.

Answer: The data were normally distributed and hence, in the study we have used parametric tests for the statistical analysis. This has been mentioned in the statistical analysis section of the manuscript as per the reviewer’s valuable suggestion. (Lines: )

Q7: The influence of gender, age, BMI, high blood pressure on the target variables periodontitis and CKD must have been tested in a multivariate analysis (e.g., logistical regression).

Answer: The reviewer’s suggestions are well taken. In the statistical analysis of the manuscript, Pearson’s correlation was done basically to correlate one variable with the other variables. The study is still in progress. In the next part of the study as per the reviewer’s suggestions the multiple logistic regression analysis will be taken into consideration and will be carried out for analysing the association between isolated systolic blood pressure, after adjusting for all the confounding factors in total. (Lines: 239-224)

Results

Q8: It is unclear whether all groups were compared with each other or each group individually with the control group.

Answer: Intergroup comparisons were carried out between the test groups and the control group, and has been explained in the results section of the manuscript as per the reviewer’s valuable suggestions. (Lines:248-251 )

Discussion and Conclusion

Q9: A statement should be issued whether Red Complex Bacteria and hypertension are a risk factor for generalized periodontitis and CKD.

Answer: A statement that red-complex bacteria and hypertension are potential risk factors for GP and CKD has been incorporated in the discussion and conclusion sections of the manuscript as per the reviewer’s valuable suggestions. (Lines:410-419)

Minor comments

Q10: A native speaker should edit the entire manuscript. Abbreviations should be explained in the text during their first use and then always applied.

Answer: Abbreviations have been expanded in the text during first use as per the reviewer’s valuable suggestions. (Lines:45)

Q11: The outline of the tables should be improved. For example, the unit of measurement for each variable should be specified. The spelling of the P values should be unified. It must emerge from the tables which groups were compared with each other and which statistical procedure was applied.

Answer: The units of measurements have been mentioned for every variable in the tables and manuscript. The spelling of p-value has been unified in the tables and manuscript.

Q12: forsythensis improve in T. forsythia.

 Answer: The spelling has been corrected in the manuscript as per the reviewer’s valuable suggestions. (Lines:86)

Q13: mMhg improve in mmHg

Answer:   The spelling has been corrected in the manuscript as per the reviewer’s valuable suggestions. (Lines:283)

Reviewer 2 Report

The paper entitled “Isolated systolic blood pressure and red-complex bacteria – a  risk for generalised periodontitis and chronic kidney disease” is an interesting original paper well written aiming to examine demographic variables, periodontal parameters, renal parameters, isolated systolic blood pressure, and the red-complex bacteria in patients with generalised periodontitis and chronic kidney disease.

The paper is well written, but in some points, it lacks clarity and correctness. Some sentences are hard to read, and I suggest breaking them into multiple sentences and/or revising the English by a native speaker or via appropriate tools, such as Grammarly. 

The abstract is complete and precise. I suggest avoiding acronyms without explanations (ISBP..RCB..)

The introduction is almost complete. I suggest adding further details about the oral dysbiosis associated with systemic disease. For this purpose, the following reference and similar could be helpful -but not mandatory- :

Contaldo, M.; et al. Overview on Osteoporosis, Periodontitis and Oral Dysbiosis: The Emerging Role of Oral Microbiota. Appl. Sci. 202010, 6000. https://doi.org/10.3390/app10176000

Contaldo M, et al. The Oral Microbiota Changes in Orthodontic Patients and Effects on Oral Health: An Overview. J Clin Med. 2021 Feb 16;10(4):780. doi: 10.3390/jcm10040780. PMID: 33669186; PMCID: PMC7919675.

Contaldo M, et al. Oral Microbiota and Salivary Levels of Oral Pathogens in Gastro-Intestinal Diseases: Current Knowledge and Exploratory Study. Microorganisms. 2021 May 14;9(5):1064. doi: 10.3390/microorganisms9051064. PMID: 34069179; PMCID: PMC8156550.

M&m section is appropriate and detailed. It allows to the reproduction of the study and sound scientific. 

In “results”, avoid repeating in the text some findings reported in tables (as the mean age ..)and prefer to refer to the tables. 

In tables, please uniform the way you wrote the numbers: sometimes it is .249..other times 0.249.. choose to use or remove the zero units everywhere. 

In Fig 1, please, use a font size larger to read the text within the figure better. 

The discussion section is the most appreciated part: it was well conducted with the proper criticism related to the study limitations and strengths. 

I recommend a light revision of the English, figure 1 adjustment as the numbers in tables. 

Author Response

Reply to Reviewer’s Comments

Reviewer 2:

The paper entitled “Isolated systolic blood pressure and red-complex bacteria – a risk for generalised periodontitis and chronic kidney disease” is an interesting original paper well written aiming to examine demographic variables, periodontal parameters, renal parameters, isolated systolic blood pressure, and the red-complex bacteria in patients with generalised periodontitis and chronic kidney disease.

Q1: The paper is well written, but in some points, it lacks clarity and correctness. Some sentences are hard to read, and I suggest breaking them into multiple sentences and/or revising the English by a native speaker or via appropriate tools, such as Grammarly. The abstract is complete and precise. I suggest avoiding acronyms without explanations (ISBP..RCB..)

Answer: The sentences have been simplified as per the reviewer’s suggestions along with the expansion of the abbreviated terms in the abstract. (Lines:45)

Q2: The introduction is almost complete. I suggest adding further details about the oral dysbiosis associated with systemic disease. For this purpose, the following reference and similar could be helpful -but not mandatory- : Contaldo, M.; et al. Overview on Osteoporosis, Periodontitis and Oral Dysbiosis: The Emerging Role of Oral Microbiota. Appl. Sci. 202010, 6000. https://doi.org/10.3390/app10176000. Contaldo M, et al. The Oral Microbiota Changes in Orthodontic Patients and Effects on Oral Health: An Overview. J Clin Med. 2021 Feb 16;10(4):780. doi: 10.3390/jcm10040780. PMID: 33669186; PMCID: PMC7919675.Contaldo M, et al. Oral Microbiota and Salivary Levels of Oral Pathogens in Gastro-Intestinal Diseases: Current Knowledge and Exploratory Study. Microorganisms. 2021 May 14;9(5):1064. doi: 10.3390/microorganisms9051064. PMID: 34069179; PMCID: PMC8156550.

Answer: Details about the oral dysbiosis associated with systemic disease has been incorporated by using the above citations as references in the introduction section of the manuscript as per the reviewer’s valuable suggestions. (Lines:94)

Q3: M&m section is appropriate and detailed. It allows to the reproduction of the study and sound scientific. 

Answer: We thank the reviewers for their encouraging comments.

Q4: In “results”, avoid repeating in the text some findings reported in tables (as the mean age ..)and prefer to refer to the tables. 

 Answer: The results section has been modified as per the reviewer’s valuable suggestion. (Lines:248-250)

Q5: In tables, please uniform the way you wrote the numbers: sometimes it is .249..other times 0.249.. choose to use or remove the zero units everywhere. 

 Answer: The values in the tables have been unified as per the reviewer’s valuable suggestion. (Lines:296,302)

Q6: In Fig 1, please, use a font size larger to read the text within the figure better. 

Answer: Larger font size has been changed in Fig 1 as per the valuable suggestions.  (Lines:144)

The discussion section is the most appreciated part: it was well conducted with the proper criticism related to the study limitations and strengths. 

Answer: We thank the reviewer for their encouraging comments.

Q7: I recommend a light revision of the English, figure 1 adjustment as the numbers in tables. 

Answer: We thank the reviewer for his/her valuable suggestions and all changes have been imported into the manuscript. (Line:144)

Round 2

Reviewer 1 Report

The authors improved their manuscript. Nevertheless, I have some questions and suggestions for improvement. In addition, the manuscript still contains many typo error.

Major concerns

Abstract

Please supplement: Control group: no CKD and no periodontitis

Methods

The authors wrote that parodontitis patient with stage II degree B were recruited. I think the authors mean that patients with periodontitis were included who have at least stage II degree B disease.

Periodontal parameters

In yellow marked text are many typos and grammatical mistakes.

I point out that the authors should describe how the plaque and gingivitis indices were determined.

Determination of PPD and CAL: How many sites per tooth were measured?

Percentage sites of pockets: Here a limit must be specified, e.g. PPD at least 4 mm.

I do not understand the periodontitis criteria (1) and (2). Generalized periodontitis for each stage means that at least 30% of the teeth had to be affected by interdental attachment loss or bone loss.

Red complex bacteria

First sentence: It has to be called: “Tannerella forsythia”

What for a transport medium was the plaque samples transferred? Since only the bacterial DNA was analyzed, no medium is actually required in which the bacteria remain vital.

Statistical analyses

Which test was applied in order to test the metric data for normal distribution?

Results

It is unusual to indicate a p-value of 0.00, better would be p <0.0001.

Tables: For all metric variables, the mean and the standard deviation (SD) should be specified. For eample: Mean age ± SD (years).

Please specify the statistical procedures used in the footnote of each table.

Table 3: Please indicate in the text that the number of missing teeth is reduced in the patient groups, but this difference is not significant.

Page 8, first sentence: please improve: Isolated SBP was statistically significant increased…

Conclusion

Please improve: “generalized periodontitis”

Minor comments

A native speaker should edit the entire manuscript. There are still many spelling and grammatical mistakes throughout the manuscript.

Author Response

REPLY TO REVIEWER’S COMMENTS

Reviewer 1:

The authors improved their manuscript. Nevertheless, I have some questions and suggestions for improvement. In addition, the manuscript still contains many typo error.

Major concerns

Abstract

Q1: Please supplement: Control group: no CKD and no periodontitis

Answer: The explanation of control group has been rectified as per reviewer’s suggestions. (line 36)

Methods

Q2: The authors wrote that periodontitis patient with stage II degree B were recruited. I think the authors mean that patients with periodontitis were included who have at least stage II degree B disease.

Answer: Thank you for your suggestions. Yes, we agree with the reviewer’s suggestion. We have recruited patients with stage II grade B periodontitis according to recent AAP classification of 2017. (lines 163-164)

Periodontal parameters

Q3: I point out that the authors should describe how the plaque and gingivitis indices were determined.

Answer: The description for plaque and gingival indices have been added and highlighted as per reviewer’s suggestions (lines 154-157)

Q4: Determination of PPD and CAL: How many sites per tooth were measured?

Answer: The description for PPD and CAL has been added and highlighted as per reviewer’s suggestions (lines 152,153)

Q5: Percentage sites of pockets: Here a limit must be specified, e.g. PPD at least 4 mm.

Answer: The description for percentage sites of pockets has been added and highlighted as per reviewer’s suggestions (line 157,158)

Q6: I do not understand the periodontitis criteria (1) and (2). Generalized periodontitis for each stage means that at least 30% of the teeth had to be affected by interdental attachment loss or bone loss.

 Answer: Reviewer’s suggestions are well taken. According to the reviewer the sentence has been modified based on the % of sites affected by attachment loss and periodontitis. (lines 160-166)

Red complex bacteria

Q7: First sentence: It has to be called: “Tannerella forsythia”

Answer: The name of the bacteria has been changed and highlighted as per reviewer’s suggestions (line 198)

Q8: What for a transport medium was the plaque samples transferred? Since only the bacterial DNA was analyzed, no medium is actually required in which the bacteria remain vital.

Answer: The sentence has been revised and modified as per reviewer’s suggestion and has been highlighted. (lines 203,204)

Statistical analyses

Q9: Which test was applied in order to test the metric data for normal distribution?

Answer: The sentence has been added and highlighted as per reviewer’s suggestions (line 241,242)

Results

Q10: It is unusual to indicate a p-value of 0.00, better would be p <0.0001.

Answer: The p-value has been modified as per reviewer’s comments

Q11: Tables: For all metric variables, the mean and the standard deviation (SD) should be specified. For eample: Mean age ± SD (years).

Answer: The Mean ± SD has been mentioned in the tables for metric variables as per reviewer’s suggestion.

Q12:Please specify the statistical procedures used in the footnote of each table.

Answer: The statistical procedure has been mentioned in the footnote of each table and highlighted.

Q13:Table 3: Please indicate in the text that the number of missing teeth is reduced in the patient groups, but this difference is not significant.

 Answer: The sentence has been revised and modified as per reviewer’s suggestion and has been highlighted. (lines 260,261)

Q14:Page 8, first sentence: please improve: Isolated SBP was statistically significant increased…

Answer: The sentence has been revised and modified as per reviewer’s suggestion and has been highlighted. (lines 285,286)

Conclusion

Q15:Please improve: “generalized periodontitis”

Answer: The term generalized periodontitis has been improved and replaced with generalized stage II grade B periodontitis as per the reviewer’s suggestions (lines 469,470)